# Community detection using fast low-cardinality semidefinite programming

**Po-Wei Wang**
Machine Learning Department
Carnegie-Mellon University
Pittsburgh, PA
poweiw@cs.cmu.edu

**J. Zico Kolter**
Department of Computer Science
Carnegie-Mellon University &
Bosch Center for Artificial Intelligence
Pittsburgh, PA
zkolter@cs.cmu.edu

## Abstract

Modularity maximization has been a fundamental tool for understanding the community structure of a network, but the underlying optimization problem is nonconvex and NP-hard to solve. State-of-the-art algorithms like the Louvain or Leiden methods focus on different heuristics to help escape local optima, but they still depend on a greedy step that moves node assignment locally and is prone to getting trapped. In this paper, we propose a new class of low-cardinality algorithm that generalizes the local update to maximize a semidefinite relaxation derived from max-k-cut. This proposed algorithm is scalable, empirically achieves the global semidefinite optimality for small cases, and outperforms the state-of-the-art algorithms in real-world datasets with little additional time cost. From the algorithmic perspective, it also opens a new avenue for scaling-up semidefinite programming when the solutions are sparse instead of low-rank.

## 1 Introduction

Community detection, that is, finding clusters of densely connected nodes in a network, is a fundamental topic in network science. A popular class of methods for community detection, called *modularity maximization* [34], tries to maximize the modularity of the cluster assignment, the quality of partitions defined by the difference between the number of edges inside a community and the expected number of such edges. However, optimizing modularity is NP-hard [14], so modern methods focus on heuristics to escape local optima. A very popular heuristic, the Louvain method [8], greedily updates the community membership node by node to the best possible neighboring community that maximizes the modularity function's gain. Then it aggregates the resulting partition and repeats until no new communities are created. The Louvain method is fast and effective [48], although it still gets trapped at local optima and might even create disconnected communities. A follow-up work, the Leiden method [43], resolves disconnectedness by an additional refinement step, but it still relies on greedy local updates and is prone to local optima.

In this paper, we propose the *Locale* (low-cardinality embedding) algorithm, which improves the performance of community detecion above the current state of the art. It generalizes the greedy local move procedure of the Louvain and Leiden methods by optimizing a semidefinite relaxation of modularity, which originates from the extremal case of the max-$k$-cut semidefinite approximation [22, 20, 2] when $k$ goes to infinity. We provide a scalable solver for this semidefinite relaxation by exploiting the *low-cardinality* property in the solution space. Traditionally, semidefinite programming is considered unscalable. Recent advances in Riemannian manifold optimization [38, 16, 1] provide a chance to scale-up by optimizing directly in a low-rank solution space, but it is not amenable in many relaxations like the max-$k$-cut SDP, where there are nonnegativity constraints on all entries of

the semidefinite variable $X$. However, due to the nonnegativity constraints, the solution $X$ is sparse and a low-cardinality solution in the factorized space $V$ suffices. These observations lead to our first contribution, which is a scalable solver for low-cardinality semidefinite programming subject to nonnegative constraints. Our second contribution is using this solver to create a generalization of existing community detection methods, which outperforms them in practice because it is less prone to local optima.

We demonstrate in the experiments that our proposed low-cardinality algorithm is far less likely to get stuck at local optima than the greedy local move procedure. On small datasets that are solvable with a traditional SDP solver, our proposed solver empirically reaches the globally optimal solution of the semidefinite relaxation given enough cardinality and is orders of magnitude faster than traditional SDP solvers. Our method uniformly improves over both the standard Louvain and Leiden methods, which are the state-of-the-art algorithms for community detection, with 2.2x time cost. Additionally, from the perspective of algorithmic design, the low-cardinality formulation opens a new avenue for scaling up semidefinite programming when the solutions tend to be sparse instead of low-rank. Source code for our implementation is available at `https://github.com/locuslab/sdp_clustering`.

## 2 Background and related work

**Notation.** We use upper-case letters for matrices and lower-case letters for vectors and scalars. For a matrix $X$, we denote the symmetric semidefinite constraint as $X \succeq 0$, the entry-wise nonnegative constraint as $X \geq 0$. For a vector $v$, we use $\mathrm{card}(v)$ for the number of nonzero entries, $\|v\|$ for the 2-norm, and $\mathrm{top}_k^+(v)$ for the sparsified vector of the same shape containing the largest $k$ nonnegative coordinates of $v$. For example, $\mathrm{top}_2^+((-1,3)) = (0,3)$, and $\mathrm{top}_1^+((-1,-2)) = (0,0)$. For a function $Q(V)$, we use $Q(v_i)$ for the same function taking the column vector $v_i$ while pinning all other variables. We use $[r]$ for the set $\{1,\ldots,r\}$, and $e(t)$ for the basis vector of coordinate $t$.

**Modularity maximization.** Modularity was proposed in [34] to measure the quality of densely connected clusters. For an undirected graph with a community assignment, its modularity is given by

$$Q(c) := \frac{1}{2m} \sum_{ij} \left[ a_{ij} - \frac{d_i d_j}{2m} \right] \delta(c_i = c_j), \tag{1}$$

where $a_{ij}$ is the edge weight connecting nodes $i$ and $j$, $d_i = \sum_j a_{ij}$ is the degree for node $i$, $m = \sum_{ij} a_{ij}/2$ is the sum of edge weights, and $c_i \in [r]$ is the community assignment for node $i$ among the $r$ possible communities. The notation $\delta(c_i = c_j)$ is the Kronecker delta, which is one when $c_i = c_j$ and zero otherwise. The higher the modularity, the better the community assignment. However, as shown in [14], optimizing modularity is NP-hard, so researchers instead focus on different heuristics, including spectral methods [33], simulated annealing [39], linear programming and semidefinite programming [2, 26]. The most popular heuristic, the Louvain method [8], initializes each node with a unique community and updates the modularity $Q(c)$ cyclically by moving $c_i$ to the best neighboring communities [27, 33]. When no local improvement can be made, it aggregates nodes with the same community and repeats itself until no new communities are created. Experiments show that it is fast and performant [48] and can be further accelerated by choosing update orders wisely [36, 4]. However, the local steps can easily get stuck at local optima, and it may even create disconnected communities [43] containing disconnected components. In follow-up work, the Leiden method [43] fixes the issue of disconnected communities by adding a refinement step that splits disconnected communities before aggregation. However, it still depends on greedy local steps and still suffers from local optima. Beyond modularity maximization, there are many other metrics to optimize for community detection, including asymptotic suprise [42], motif-aware [30] or higher order objectives [49].

**Semidefinite programming and clustering.** Semidefinite programming (SDP) has been a powerful tool in approximating NP-complete problems [28, 37]. Specifically, the max-$k$-cut SDP [22, 20] deeply relates to community detection, where max-$k$-cut maximizes the sum of edge weights *between* partitions, while community detection maximizes the sum *inside* partitions. The max-$k$-cut SDP is given by the optimization problem

$$\underset{X}{\mathrm{maximize}} \quad -\sum_{ij} a_{ij} x_{ij}, \text{ s.t. } X \succeq 0, \ X \geq -1/(k-1), \ \mathrm{diag}(X) = 1. \tag{2}$$

When limiting the rank of $X$ to be $k-1$, values of $x_{ij}$ become discrete and are either 1 or $-1/(k-1)$, which works similarly to a Kronecker delta $\delta(c_i = c_j)$ [20]. If $k$ goes to infinity, the constraint set reduces to $\{X \geq 0, X \succeq 0, X_{ii} = 1\}$, and Swamy [41] provides a 0.75 approximation ratio to correlation clustering on the relaxation (the bound doesn't apply to modularity maximization). However, these SDP relaxations are less practical because known semidefinite solvers don't scale with the numerous constraints, and the runtime of the rounding procedure converting the continuous variables to discrete assignments grows with $O(n^2 k)$. By considering max-2-cut iteratively at every hierarchical level, Agarwal and Kempe [2] is able to perform well on small datasets by combining SDPs with the greedy local move, but the method is still unscalable due to the SDP solver. DasGupta and Desai [17] discussed the theoretical property of SDPs when there are only 2 clusters. Other works [44, 26] also apply SDPs to solve clustering problems, but they don't optimize modularity.

**Low-rank methods for SDP.** One trick to scale-up semidefinite programming is to exploit its low-rank structure when possible. The seminal works by Barvinok and Pataki [38, 6] proved that when there are $m$ constraints in an SDP, there must be an optimal solution with rank less than $\sqrt{2m}$, and Barvinok [5] proved the bound to be tight. Thus, when the number of constraints $m$ is subquadratic, we can factorize the semidefinite solution $X$ as $V^T V$, where the matrix $V$ requires only $n\sqrt{2m}$ memory instead of the original $n^2$ of $X$. Burer and Monteiro [16] first exploited this property and combined it with L-BFGS to create the first low-rank solver of SDPs. Later, a series of works on Riemannian optimization [1, 12, 11, 45] further established the theoretical framework and extended the domain of applications for the low-rank optimization algorithms. However, many SDPs, including the max-k-cut SDP approximation that we use in this paper, have entry-wise constraints on $X$ like $X \geq 0$, and thus the low-rank methods is not amenable to those problems.

**Copositive programming.** The constraint $DNN = \{X \mid X \succeq 0, \ X \geq 0\}$ in our SDP relaxation is connected to an area called "copositive programming" [31, 19, 15], which focuses on the sets

$$CP = \{X \mid v^T X v \geq 0, \ \forall v \geq 0\} \quad \text{and} \quad CP^* = \{V^T V \mid V \geq 0, \ V \in \mathbb{R}^{n \times r}, \ \forall r \in \mathbb{N}\}. \tag{3}$$

Interestingly, both $CP$ and $CP^*$ are convex, but the set membership query is NP-hard. The copositive sets relate to semidefinite cone $S = \{X \mid v^T X v \geq 0, \ \forall v\}$ by the hierarchy

$$CP \supseteq S \supseteq DNN \supseteq CP^*. \tag{4}$$

For low dimensions $n \leq 4$, the set $DNN = CP^*$, but $DNN \supsetneq CP^*$ for $n \geq 5$ [23]. Optimization over the copositive set is hard in the worst case because it contains the class of binary quadratic programming [15]. Approximation through optimizing the $V$ space has been proposed in [9, 24], but it is still time-consuming because it requires a large copositive rank $r = O(n^2)$ [10].

## 3 The Locale algorithm and application to community detection

In this section, we present the Locale (low-cardinality embedding) algorithm for community detection, which generalizes the greedy local move procedure from the Louvain and Leiden methods. We describe how to derive the low-cardinality embedding from the local move procedure, its connection to the semidefinite relaxation, and then how to round the embedding back to the discrete community assignments. Finally, we show how to incorporate this algorithm into full community detection methods.

### 3.1 Generalizing the local move procedure by low-cardinality embeddings

State-of-the-art community detection algorithms like the Louvain and Leiden methods depend on a core component, the local move procedure, which locally optimizes the community assignment for a node. It was originally proposed by Kernighan and Lin [27] for graph cuts, and was later adopted by Newman [33] to maximize the modularity $Q(c)$ defined in (1). The local move procedure in [33, 8] first initializes each node with a unique community, then updates the community assignment node by node and changes $c_i$ to a neighboring community (or an empty community) that maximizes the increment of $Q(c_i)$. That is, the local move procedure is an *exact coordinate ascent method* on the discrete community assignment $c$. Because it operates on the discrete space, it is prone to local optima. To improve it, we will first introduce a generalized maximum modularity problem such that each node may belong to more than one community.

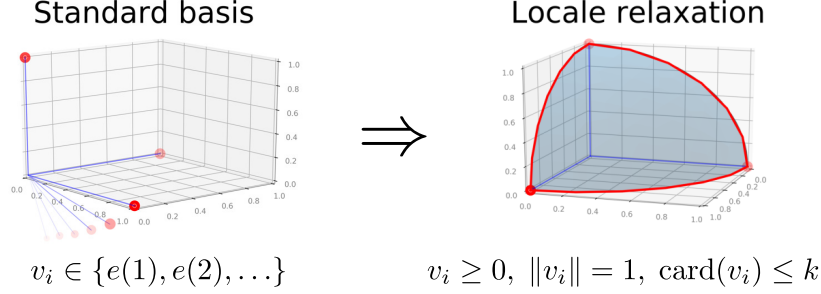

| Standard basis | Locale relaxation |
|---|---|
| $v_i \in \{e(1), e(2), \dots\}$ | $v_i \geq 0,\ \|v_i\| = 1,\ \mathrm{card}(v_i) \leq k$ |

Figure 1: An illustration of the low cardinality relaxation, where the discrete cluster assignment for nodes is relaxed into a continuous and smooth space containing the original discrete set. The parameter $k$ controls the cardinality, or equivalently the maximum number of overlapping communities a node may belong to. When $k = 1$, we recover the original discrete set.

**A generalized maximum modularity problem.** To assign a node to more than one community, we need to rewrite the Kronecker delta $\delta(c_i = c_j)$ in $Q(c)$ as a dot product between basis vectors. Let $e(t)$ be the basis vector for community $t$ with one in $e(t)_t$ and zeros otherwise. By creating an assignment vector $v_i = e(c_i)$ for each node $i$, we have $\delta(c_i = c_j) = v_i^T v_j$, and we reparameterize the modularity function $Q(c)$ defined in (1) as

$$Q(V) := \frac{1}{2m} \sum_{ij} \left[ a_{ij} - \frac{d_i d_j}{2m} \right] v_i^T v_j. \qquad (5)$$

Notice that the original constraint $c_i \in [r]$, where $r$ is the upper-bounds on number of communities, becomes $v_i \in \{e(t) \mid t \in [r]\}$. And the set becomes equivalent to the below unit norm and unit cardinality constraint in the nonnegative orthant.

$$\{e(t) \mid t \in [r]\} = \{v_i \mid v_i \in \mathbb{R}_+^r,\ \|v_i\| = 1,\ \mathrm{card}(v_i) \leq 1\}. \qquad (6)$$

The constraint can be interpreted as the intersection between the curved probability simplex ($v_i \in \mathbb{R}_+^r,\ \|v_i\| = 1$) and the cardinality constraint ($\mathrm{card}(v_i) \leq 1$), where the latter constraint controls how many communities may be assigned to a node. Naturally, we can generalize the maximum modularity problem by relaxing the cardinality constraint from $1$ to $k$, where $k$ is the maximun number of overlaying communities a node may belong to. The generalized problem is given by

$$\underset{V}{\text{maximize}}\ Q(V) := \frac{1}{2m} \sum_{ij} \left[ a_{ij} - \frac{d_i d_j}{2m} \right] v_i^T v_j,\quad \text{s.t. } v_i \in \mathbb{R}_+^r,\ \|v_i\| = 1,\ \mathrm{card}(v_i) \leq k,\ \forall i. \qquad (7)$$

The larger the $k$, the smoother the problem (7). When $k = r$ the cardinality constraint becomes trivial and the feasible space of $V$ become smooth. The original local move procedure is simply an exact coordinate ascent method when $k = 1$, and we now generalized it to work on arbitrary $k$ in a smoother feasible space of $V$. We call the resulting $V$ the low cadinality embeddings and the generalized algorithm the *Locale* algorithm. An illustration is given in Figure 1.

**The Locale algorithm for low-cardinality embeddings.** We first prove that, just like the local move procedure, there is a closed-form optimal solution for the subproblem $Q(v_i)$, where we optimize on variable $v_i$ and pin all the other variables.

**Proposition 1.** *The subproblem for variable* $v_i$

$$\underset{v_i}{\text{maximize}}\ Q(v_i),\quad \text{s.t. } v_i \in \mathbb{R}_+^r,\ \|v_i\| = 1,\ \mathrm{card}(v_i) \leq k \qquad (8)$$

*admits the following optimal solution*

$$v_i = g/\|g\|,\quad \text{where } g = \begin{cases} e(t) \text{ with the max } (\nabla Q(v_i))_t & \text{if } \nabla Q(v_i) \leq 0 \\ top_k^+(\nabla Q(v_i)) & \text{otherwise} \end{cases}, \qquad (9)$$

*where* $top_k^+(q)$ *is the sparsified vector containing the top-k-largest nonnegative coordinates of q. For the special case* $\nabla Q(v_i) \leq 0$*, we choose the* $t$ *with maximum* $(v_i)_t$ *from the previous iteration if there are multiple* $t$ *with maximum* $(\nabla Q(v_i))_t$*.*

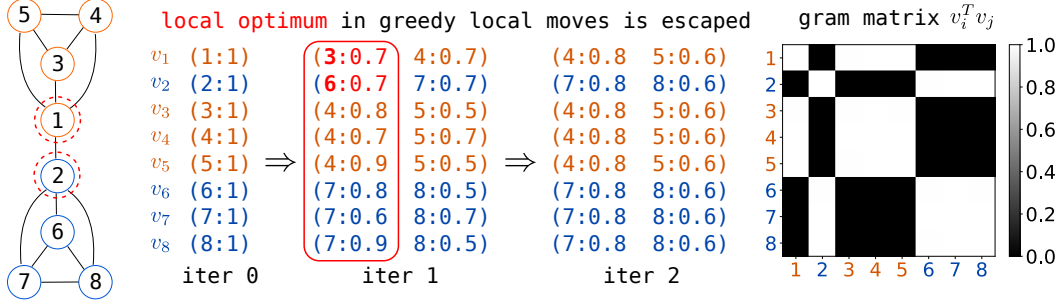

Figure 2: An example that the Locale algorithm escapes the local optimum in greedy local move procedure. Numbers in the parentheses are the low-cardinality embeddings in a sparse `index:value` format, where we compress a sparse vector with its top-$k$ nonzero entries. The above bottleneck graph was used in the Leiden paper [43] to illustration local optima, where a greedy local move procedure following the order of the nodes gets stuck at the local optima in the red box, splitting node 1 and 2 from the correct communities because of its unit cardinality constraint. In contrast, the Locale (low-cardinality embeddings) algorithm escapes the local optima because it has an additional channel for the top-$k$ communities to cross the bottleneck. The gram matrix of the resulting embeddings shows that it perfectly identifies the communities.

We list the proof in Appendix A. With the close-form solution for every subproblem, we can now generalize the local move procedure to a low-cardinality move procedure that works on arbitrary $k$. We first initialize every vector $v_i$ with a unique vector $e(i)$, then perform the *exact block coordinate ascent* method using the optimal update (9) cycling through all variables $v_i$, $i = 1, \ldots, n$, till convergence. We could also pick coordinate randomly, and because the updates are exact, we have the following guarantee.

**Theorem 2.** *Applying the low-cardinality update iteratively on random coordinates[1], the projected gradient of the iterates converges to zero at $O(1/T)$ rate, where $T$ is the number of iterations.*

We list the proof in Appendix B. When implementing the Locale algorithm, we store the matrix $V$ in a sparse format since it has a fixed cardinality upper bound, and perform all the summation using sparse vector operations. We maintain a vector $z = \sum_j d_j v_j$ and compute $\nabla Q(v_i)$ by $(\sum_j a_{ij} v_j) - \frac{d_i}{2m}(z - d_i v_i)$. This way, updating all $v_i$ once takes $O(\mathrm{card}(A) \cdot k \log k)$ time, where the $\log k$ term comes from the partial sort to implement the $\mathrm{top}_k^+(\cdot)$ operator. Taking a small $k$ (we pick $k = 8$ in practice), the experiments show that it scales to large networks without too much additional time cost to the greedy local move procedure. Implementation-wise, we choose the updating order by the smart local move [36, 4]. We initialize $r$ to be the number of nodes and increase it when $\nabla Q(v_i) \leq 0$ and there is no free coordinate. This corresponds to the assignment to a new "empty community" in the Louvain method [8] (which also increases the $r$). At the worst case, the maximum $r$ is $n \cdot k$, but we have never observed this in the experiments, where in practice $r$ is always less than $2n$. For illustration, we provide an example from Leiden method [43] in Figure 2 showing that, because of the relaxed cardinality constraint, the Locale algorithm is less likely to get stuck at local optima compared to the greedy local move procedure.

**Connections to correlation clustering SDP and copositive programming.** Here we connect the Locale solution to an SDP relaxation of the generalized modularity maximization problem (7). Let $r$ to be large enough[2] and let $k = r$ to drop the cardinality constraint, the resulting feasible gram matrix of $V$ becomes (the dual of) the copositive constraint

$$\{V^T V \mid V \geq 0, \ \mathrm{diag}(V^T V) = 1\}, \tag{10}$$

which can be further relaxed to the semidefinite constraint

$$\{X \mid X \succeq 0, \ X \geq 0, \ \mathrm{diag}(X) = 1\}. \tag{11}$$

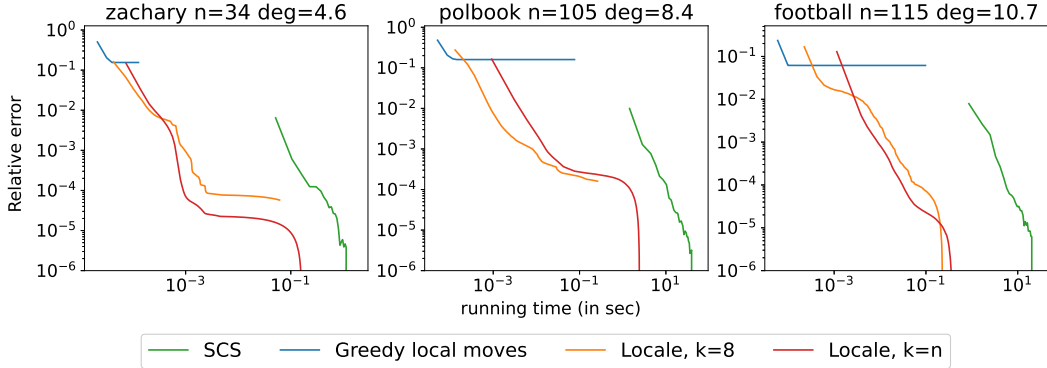

Figure 3: Comparing the relative error to optimal objective values and the running time in the semidefinite relaxation of maximum modularity. The optimal values are obtained by running the SCS [35], a splitting conic solver, for 3k iterations. The greedy local move procedure gets stuck pretty early at a local optimum (even for the original modularity maximization problem). The Locale algorithm is able to give a good approximation with cardinality $k = 8$, and is able to reach the global optima with $k = n$. Further, it is 100 to 1000 times faster than SCS, which is already orders of magnitude faster the than state-of-the-art interior point methods.

This semidefinite constraint has been proposed as a relaxation for correlation clustering in [41]. With these relaxations, the complete SDP relaxation for the (generalized) maximum modularity problem is

$$\underset{X}{\text{maximize}} \quad \sum_{ij} \left[ a_{ij} - \frac{d_i d_j}{2m} \right] x_{ij}, \quad \text{s.t. } X \succeq 0, \ X \geq 0, \ \text{diag}(X) = 1. \tag{12}$$

We use the SDP relaxation to certify whether the Locale algorithm reaches the global optima, given enough cardinality $k$. That is, if the objective value given by the Locale algorithm meets the SDP relaxation (solvable via an SDP solver), it certifies the globally optimality of (7). In the experiments for small datasets that is solvable via SDP solvers, we show that a very low cardinality $k = 8$ is enough to approximate the optimal solution to a difference of $10^{-4}$, and running the Locale algorithm with $k = n$ recovers the global optimum. In addition, our algorithm is orders-of-magnitude faster than the state-of-the-art SDP solvers.

## 3.2 Rounding by changing the cardinality constraint

After obtaining the embeddings for the generalized modularity maximization algorithm (7), we need to convert the embedding back to unit cardinality to recover the community assignment for the original maximum modularity problem. This is achieved by running the Locale algorithm with the $k = 1$ constraint, starting at the previous solution. Also, since the rounding procedure reduces all embeddings to unit cardinality after the first sweep, this is equivalent to running the local move procedure of the Louvain method, but initialized with higher-cardinality embeddings. Likewise, we could also increase the cardinality constraint to update a unit cardinality solution to a higher cardinality solution. These upgrade and downgrade steps can be performed iteratively to increase the quality of the solution slowly, but we find that it is more efficient to only do the downgrade steps in the overall multi-level algorithm. The rounding process has the same complexity as the Locale algorithm since it is a special case of the algorithm with $k = 1$.

## 3.3 The Leiden-Locale algorithm for community detection

Here, we assemble all the aforementioned components and build the Leiden-Locale algorithm for community detection. We use the Leiden method [43] as a framework and replace the local move procedure with the Locale algorithm followed by the rounding procedure. While the results are better with more inner iterations of the Locale algorithm, we found that two inner iterations followed by the rounding procedure is more efficient in the overall multi-level algorithm over multiple iterations, while substantially improving over past works. We list the core pseudo-code below, and the subroutines can be found in the Appendix E.

---

**Algorithm 1** The Leiden-Locale method

---

 1: **procedure** LEIDEN-LOCALE(Graph $G$, Partition $P$)
 2:     **do**
 3:         $E \leftarrow$ `LocaleEmbeddings`$(G, P)$                ▷ Replace the `LocalMove`$(G, P)$ in Leiden
 4:         $P \leftarrow$ `LocaleRounding`$(G, P, E)$
 5:         $G, P,$ done $\leftarrow$ `LeidenRefineAggregate`$(G, P)$        ▷ [43, Algorithm A.2, line 5-9]
 6:     **while** not done
 7:     **return** $P$
 8: **end procedure**

---

Because we still use the refinement step from the Leiden algorithm, we have the following guarantee.

**Theorem 3.** *[43, Thm. 5] The communities obtained from the Leiden-Locale algorithm are connected.*

Since we only perform two rounds of updates of the Locale algorithm, it adds relatively little overhead and complexity to the Leiden algorithm. However, experiments show that the boost is significant. The Leiden-Locale algorithm gives consistently better results than the other state-of-the-art methods.

## 4    Experimental results

In this section, we evaluate the Locale algorithm with other state-of-the-art methods. We show that the Locale algorithm is effective on the semidefinite relaxation of modularity maximization, improves the complexity from $O(n^6)$ to $O(\text{card}(A)k \log k)$ over SDP solvers, and scales to millions of variables. Further, we show that on the original maximum modularity problem, the Locale algorithm significantly improves the greedy local move procedure. When used on the community detection problem, the combined Leiden-Locale algorithm provides a 30% additional performance increase over ten iterations and is better than all the state-of-the-art methods on the large-scale datasets compared in the Leiden paper [43], with 2.2x the time cost to the Leiden method. The code for the experiment is available at the supplementary material.

**Comparison to SDP solvers on the semidefinite relaxation of maximum modularity.**    We compare the Locale algorithm to the state-of-the-art SDP solver on the semidefinite relaxation (12) of the maximum modularity problem. We show that it converges to the global optimum of SDP on the verifiable datasets and is much faster than the SDP method. We use 3 standard toy networks that are small enough to be solvable via an SDP solver, including `zachary` [50], `polbook` [34], and `football` [21]. Typically, primal-dual interior-point methods [40] have cubic complexity in the number of variables, which is $n^2$ in our problem, so the total complexity is $O(n^6)$. Moreover, the canonical SDP solver requires putting the nonnegativity constraint on the diagonal of the semidefinite variable $X$, leading a much higher number of variables. For fairness, we choose to compare with a new splitting conic solver, the SCS [35], which supports splitting variables into Cartesian products of cones, which is much more efficient than pure SDP solver in this kind of problem. We run the SCS solver for 3k iterations for the reference optimal objective value. Also, since the solution of SCS might not be feasible, we project the solution back to the feasible set by iterative projections.

Figure 3 shows the plot of difference to the optimal objective value and the running time. The greedy local move procedure gets stuck at a local optimum early in the plot, but the Locale algorithm gives a decent approximation at a low cardinality $k = 8$. At $k = n$, both the Locale algorithm and the SCS solver reach the optimum for the SDP relaxation (12), but Locale is 193x faster than SCS (in average) for reaching $10^{-4}$ difference to the optimum, and the speedup scales with the dimensions.

The results demonstrate the effectiveness and the orders of speedup of the proposed low-cardinality method, opening a new avenue for scaling-up semidefinite programming when the solution is low-cardinality instead of low-rank.

**Comparison with the local move procedure.**    In this experiment, we show that the Locale algorithm scales to millions of nodes and significantly improves the local move procedure in empirical networks. We compare to 5 large-scale networks, including DBLP, `Amazon`, `Youtube` [47], IMDB[46],and `Live Journal` [3, 29]. These are the networks that were also studied in the Leiden [43] and Louvain papers [8]. For the greedy local move procedure, we run it iteratively till convergence (or till the

Table 1: Overview of the empirical networks and the modularity after the greedy local move procedure (running till convergence) and the Locale algorithm (running for 2 rounds or till convergence).

| Dataset | Nodes | Degree | Greedy local moves | The Locale algorithm 2 rounds | full update |
|---------|-------|--------|--------------------|-----------------|-------------|
| DBLP | 317 080 | 6.6 | 0.5898 | 0.6692 | 0.8160 |
| Amazon | 334 863 | 5.6 | 0.6758 | 0.7430 | 0.9154 |
| IMDB | 374 511 | 80.2 | 0.6580 | 0.6697 | 0.6852 |
| Youtube | 1 134 890 | 5.3 | 0.6165 | 0.6294 | 0.7115 |
| Live Journal | 3 997 962 | 17.4 | 0.6658 | 0.6540 | 0.7585 |

function increment is less than $10^{-8}$ after $n$ consecutive changes to avoid floating-point errors). For the Locale algorithm, we test two different settings: running only two rounds of updates or running it till convergence, followed by the rounding procedure.

Table 1 shows the comparison results. With only 2 rounds of updates, the Locale algorithm already improves the greedy local move procedure except for the `Live Journal` dataset. When running a full update, the algorithm significantly outperforms the greedy local moves on all datasets. Moreover, it is even comparable with running a full (multi-level) iteration of the Louvain and Leiden methods, as we will see in the next experiments. The results suggest that the Locale algorithm indeed improves the greedy local move procedure. With the algorithm, we create a generalization of the Leiden method and show that it outperforms the Leiden methods in the next experiment.

**Comparison with state-of-the-art community detection algorithms.** In the experiments, we show that the Leiden-Locale algorithm (Locale in the table) outperforms the Louvain and Leiden methods, the state-of-the-art community detection algorithms. For more context, the Louvain and Leiden methods are the state-of-the-art multi-level algorithm that performs the greedy local move procedure, refinement (for Leiden only), and aggregation of graph at every level until convergence. Further, they can be run for multiple iterations using previous results as initialization[3], so we consider the settings of running the algorithm once or for 10 iterations, where one iteration means running the whole multi-level algorithm until convergence. Specifically, for the 10-iteration setting, we take the best results over 10 random trials for the Louvain and Leiden methods shown in the Leiden paper [43]. Since the Locale algorithm is less sensitive to random seeds, we only need to run it once in the setting. This make it much faster than the Leiden method with 10 trials, while performing better. Further, we run the inner update twice for the Locale algorithm since we found that it is more efficient in the overall multi-level algorithm over multiple iterations.

Table 2 shows the result of comparisons. In the one iteration setting, the Locale algorithm outperforms both the Louvain and Leiden methods (except for the `Live Journal` dataset), with an 2.2x time cost in average. Using the Louvain method as a baseline, the Locale method provides a 0.0052 improvement in average and the Leiden method provides a 0.0034 improvement. These improvement are significant since little changes in modularity can give completely different community assignments [2]. For the 10 iteration setting, we uniformly outperform both Louvain and Leiden methods in all datasets and provide a 30% additional improvement over the Leiden method using the Louvain method as a baseline.

## 5   Conclusion

In this paper, we have presented the Locale (low-cardinality embeddings) algorithm for community detection. It generalizes the greedy local move procedure from the Louvain and Leiden methods by approximately solving a low-cardinality semidefinite relaxation. The proposed algorithm is scalable (orders of magnitude faster than the state-of-the-art SDP solvers), empirically reaches the global optimum of the SDP on small datasets that we can verify with an SDP solver. Furthermore, it improves the local move update of the Louvain and Leiden methods and outperforms the state-of-the-art community detection algorithms.

Table 2: Overview of the empirical networks and the maximum modularity, running for 1 iterations (running the full multi-level algorithm till no new communities are created) and for 10 iterations (using results from the previous iteration as initialization). Note that results of Louvain [8] and Leiden [43] methods are obtained in additional 10 random trials.

| Dataset | Nodes | Degree | 1 iter | | | 10 iters 10 trials [43] | | 10 iters |
| | | | Louvain | Leiden | **Locale** | Louvain | Leiden | **Locale** |
|---|---|---|---|---|---|---|---|---|
| DBLP | 317 080 | 6.6 | 0.8201 | 0.8206 | 0.8273 | 0.8262 | 0.8387 | 0.8397 |
| Amazon | 334 863 | 5.6 | 0.9261 | 0.9252 | 0.9273 | 0.9301 | 0.9341 | 0.9344 |
| IMDB | 374 511 | 80.2 | 0.6951 | 0.7046 | 0.7054 | 0.7062 | 0.7069 | 0.7070 |
| Youtube | 1 134 890 | 5.3 | 0.7179 | 0.7254 | 0.7295 | 0.7278 | 0.7328 | 0.7355 |
| Live Journal | 3 997 962 | 17.4 | 0.7528 | 0.7576 | 0.7531 | 0.7653 | 0.7739 | 0.7747 |

The Locale algorithm can also be interpreted as solving a generalized modularity problem (7) that allows assigning at most $k$ communities to each node, and this may be intrinsically a better fit for practical use because in a social network, a person usually belongs to more than one community. Further, the Locale algorithm hints a new way to solve heavily constrained SDPs when the solution is sparse but not low-rank. It scales to millions of variables, which is well beyond previous approaches. From the algorithmic perspective, it also opens a new avenue for scaling-up semidefinite programming by the low-cardinality embeddings.

## 6 Broader impact

Although most of the work focuses on the mathematical notion of modularity maximization, the community detection algorithms that result from these methods have a broad number of applications with both potentially positive and negative benefits. Community detection methods have been used extensively in social networks (e.g., [18]), where they can be used for advertisement, tracking, attribute learning, or recommendation of new connections. These may have positive effects for the networks, of course, but as numerous recent studies also demonstrate potential negative consequences of such social network applications [13]. In these same social networks, there are also of course many positive applications of community detection algorithms. For example, researchers have used community detection methods, including the Louvain method, to detect bots in social networks [7], an activity that can bring much-needed transparency to the interactions that are becoming more common. While we do not explore such applications here, it is possible that the multi-community methods we discuss can also have an impact on the design of these methods, again for both positive or negative effects.

Ultimately, the method we present here does largely on community detection from a purely algorithmic perspective, focusing on the modularity maximization objective, and provides gains that we believe can improve the quality of existing algorithms as a whole. Ultimately, we do believe that presenting these algorithms publicly and evaluating them fairly, we will at least be able to better establish the baseline best performance that these algorithms can achieve. In other words, we hope to separate the algorithmic goal of modularity maximization (which our algorithm addresses), which is an algorithmic question, from the more applied question of what can be done with the clusters assigned by this "best available" modularity maximization approach. Specifically, if we *could* achieve the true maximum modularity community assignment for practical graphs, what would this say about the resulting communities or applications? We hope to be able to study this in future work, as our approach and others push forward the boundaries on how close we can get to this "best" community assignment.

## Acknowledgments

Po-Wei Wang is supported by a grant from the Bosch Center for Artificial Intelligence.

## Footnotes

[1] The proof can also be done with a cyclic order using Lipschitz continuity, but for simplicity we focus on the randomized version in our proof, which contains largely the same arguments and intuition.

[2] At the worse case $r = n(n+1)/2 - 4$ suffices [10, Theorem 4.1].

[3]The performance with low-number of iterations is also important because the Leiden method takes a default of 2 iterations in the `leidenalg` package and 10 iterations in the paper [43].

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
