[Supplementary Material]

# A  Proof of Proposition 1

Note that in problem (7) the term $(a_{ii} - d_i d_i/(2m))v_i^T v_i$ is constant because $\|v_i\| = 1$. Thus, in the subproblem $Q(v_i)$ for variable $v_i$, we can ignore the constant term and write the gradient $\nabla Q(v_i)$ as

$$\nabla Q(v_i) = \frac{1}{2m} \sum_{j \neq i} \left(a_{ij} - \frac{d_i d_j}{2m}\right) v_j. \tag{13}$$

Further, since there is no $v_i$ term in $\nabla Q(v_i)$, the objective function $Q(v_i)$ for the subproblem of variable $v_i$ becomes $q^T v_i$ with $q = \nabla Q(v_i)$, up to a constant. For simplicity, denote $v_i$ as $v$, and the subproblem reduces to

$$\underset{v}{\text{maximize}} \ q^T v, \ \ \text{s.t. } v \in \mathbb{R}_+^r, \ \|v\| = 1, \ \text{card}(v) \leq k. \tag{14}$$

Let $v^*$ be the optimal solution of the above subproblem (8) (existence by compactness). When $q \leq 0$, we have $\max(q) \leq 0$. With $\|v\|_2 = 1$, $v \geq 0$, and $\|v\|_2 \leq \|v\|_1$, there is

$$\max(q) = \max(q)\|v\|_2 \geq \max(q)\|v\|_1 = \max(q) \sum_t v_t \geq \sum_t q_t v_t = q^T v. \tag{15}$$

Thus, $e(t)$ with the max $q_t$ is the optimal solution in the first case. For the second case, there is at least one coordinate $p$ such that $q_p > 0$. Now we exclude the following two cases of inactive coordinates by contradictions.

**(When $q_t < 0$)**   We know $v_t^* = 0$. Otherwise, suppose there is a $v_t^* > 0$ with $q_t < 0$.

If $q^T v^* \leq 0$, selecting $v^* = e(p)$ violates the optimality of $v^*$, a contradiction.

If $q^T v^* > 0$, we have

$$0 < q^T v^* < q^T (v^* - e(t)v_t^*) \leq q^T (v^* - e(t)v_t^*)/\|v^* - e(t)v_t^*\|, \tag{16}$$

also a contradiction to the optimality of $v^*$, because the last term is a feasible solution.

**(When $q_t < q_{[k]}$, where $q_{[k]}$ is the $k$-th largest value)**   We know $v_t^* = 0$. Otherwise, there must be a coordinate $j$ in the top-$k$-largest value that is not selected ($v_j^* = 0$) because $\text{card}(v^*) \leq k$. This way, we have

$$q^T v^* < q^T (v^* - e(t)v_t^* + e(j)v_t^*), \tag{17}$$

which contradicts to the optimality of $v^*$ because $(v^* - e(t)v_t^* + e(j)v_t^*)$ is a feasible solution.

Thus, by removing the inactive coordinates, the effective objective function $q^T v^*$ becomes $\text{top}_k^+(q)^T v^*$, and the optimal solution follows from $\|v_i^*\| = 1$ and $\text{top}_k^+(q) \geq 0$. □

# B   Proof of Theorem 2

Define the projected gradient (for maximization) as

$$\texttt{grad}(V) = P_\Omega(V + \nabla Q(V)) - V, \tag{18}$$

where $P_\Omega$ is the projection (under 2-norm) to the constraint set $\Omega$ of the optimization problem (7)

$$\Omega = \{V \mid v_i \in \mathbb{R}_+^r, \ \|v_i\| = 1, \ \text{card}(v_i) \le k, \ \forall i = 1, \ldots, n\}, \tag{19}$$

and denote $\Omega_i$ as the constraint for $v_i$ for the separable $\Omega$. Because the cardinality constraint is an union between finite hyperplanes, it is a closed set, which implies the constraint of the optimization problem is a compact set. Thus, by the Weierstrass extreme value theorem, the function $Q(V)$ is upper-bounded and must attain global maximum over the constraint.

Now we connect the exact update in the Locale algorithm with the projected gradient. Denote $v_i^+$ as the update taken for the subproblem $Q(v_i)$. Because the Locale algorithm performs an exact update (Proposition 1), we have

$$\nabla Q(v_i)^T v_i^+ \ge \nabla Q(v_i)^T u, \ \forall u \in \Omega_i. \tag{20}$$

Further, because $\|v_i^+\|^2 = 1$ and $\|u\|^2 = 1$, we have

$$\|v_i^+ - \nabla Q(v_i)\|^2 \le \|u - \nabla Q(v_i)\|^2, \ \forall u \in \Omega_i. \tag{21}$$

This means that the update $v_i^+$ is the projection of $\nabla Q(v_i)$ to the constraint set $\Omega_i$. To connect the update with the projected gradient, we need the following lemma.

**Lemma 4.** *Denote the projection (under 2-norm) of a point $x$ on a closed constraint set $\Omega$ as $P_\Omega(x)$. Then for any scalar $\alpha > 1$ and vector $q$, we have*

$$q^T(P_\Omega(x + \alpha q) - P_\Omega(x + q)) \ge 0$$

The proof is listed in Appendix C. Taking the lemma with $\alpha \to \infty$ and let $q = \nabla Q(v_i)$, we have

$$0 \le \lim_{\alpha \to 0} q^T(P_{\Omega_i}(v_i + \alpha q) - P_{\Omega_i}(x + q)) = q^T(v_i^+ - P_{\Omega_i}(v_i + q)), \tag{22}$$

where the last equation follows because $v_i^+$ is the projection of $q$ on $\Omega_i$ with $\|\cdot\| = 1$ constraint [4]. Further, apply the definition of projection $P_{\Omega_i}(v_i + q)$ again on the feasible $v_i$, we have

$$\|P_{\Omega_i}(v_i + q) - (v_i + q)\|^2 \le \|v_i - (v_i + q)\|^2, \tag{23}$$

and after rearranging there is

$$\|P_{\Omega_i}(v_i + q) - v_i\|^2 \le 2q^T(P_{\Omega_i}(v_i + q) - v_i). \tag{24}$$

Applying (22) to the equation above, we have

$$\|P_{\Omega_i}(v_i + q) - v_i\|^2 \le 2q^T(v_i^+ - v_i). \tag{25}$$

The right hand side of the above equation equals the function increment $Q(v_i^+) - Q(v_i)$. Thus,

$$\|P_{\Omega_i}(v_i + q) - v_i\|^2 \le 2(Q(v_i^+) - Q(v_i)). \tag{26}$$

Now, taking expectation over the random coordinate $i$, we have

$$\frac{1}{n}\|P_\Omega(V + \nabla Q(V)) - V\|^2 = \mathbb{E}\|P_{\Omega_i}(v_i + q) - v_i\|^2 \le 2\mathbb{E}(Q(v_i^+) - Q(v_i)) = Q(V^{t+1}) - Q(V^t). \tag{27}$$

Further, since $Q(V^{t+1}) - Q(V^t)$ is monotonic increasing, summing them over iterations 0 to $T - 1$ forms a telescoping sum, which is upper-bounded by $Q(V^*) - Q(V^0)$, where $V^*$ is the global optimal solution of $Q(V)$. Substitute the definition of projected gradient (18), we have

$$\frac{T}{n} \min_t \|\texttt{grad}(V^t)\|^2 \le \frac{1}{n} \sum_{t=0}^{T-1} \|\texttt{grad}(V^t)\|^2 \le 2(Q(V^*) - Q(V^0)). \tag{28}$$

Thus, the projected gradient $\texttt{grad}(V)$ converges to zero at a $O(1/T)$ rate.   $\square$

# C Proof for Lemma 4

By definition of the projection $P_\Omega(x + q)$, we have

$$\|P_\Omega(x + q) - (x + q)\|^2 \leq \|P_\Omega(x + \alpha q) - (x + q)\|^2.$$

Take out the $q$ term out of the norm and rearrange, there is

$$\|P_\Omega(x + q) - x\|^2 \leq \|P_\Omega(x + \alpha q) - x\|^2 - 2q^T(P_\Omega(x + \alpha q) - P_\Omega(x + q)). \tag{29}$$

Similarly, by definition of the projection $P_\Omega(x + \alpha q)$, there is

$$\|P_\Omega(x + \alpha q) - x\|^2 \leq \|P_\Omega(x + q) - x\|^2 - 2\alpha q^T(P_\Omega(x + q) - P_\Omega(x + \alpha q)). \tag{30}$$

Sum (29) and (30), the norms cancel, and we have

$$2(\alpha - 1)q^T(P_\Omega(x + \alpha q) - P_\Omega(x + q)) \geq 0,$$

which implies

$$q^T(P_\Omega(x + \alpha q) - P_\Omega(x + q)) \geq 0. \tag{31}$$

Thus, the result holds. $\square$

# D Experiments on networks with ground truth

In this section, we compare results from the Leiden-Locale method on data with the ground truth for partitions. The result is listed in Figure 4.

(a) `zachary` (ground truth = 4 clusters)    (b) `polbook` (ground truth = 3 clusters)

Figure 4: The comparison of the results from Leiden-Locale method to ground-truth partitions in the `zachary` and `polbook` datasets. The position of each node is arranged using the 2D Fruchterman-Reingold force-directed algorithm from the ground-truth using `networkx` [25], and the color of each node indicates the solution community given by Leiden-Locale algorithm. The red edges between nodes indicates the case when two nodes are inside the same cluster in the ground truth but wasn't assigned so in our algorithm. For `zachary`, the Leiden-Locale algorithm returns a perfect answer comparing to the ground truth with a perfect modularity of 0.4197 [32]. For `polbook`, it misclassifies 18 over 105 nodes, but still attains a best known modularity of 0.5272 [2].

# E  Pseudo-code for the Leiden-Locale algorithm

Here we list the pseudo-code for the Leiden-Locale method. Note that we reuse Algorithm 2 in Algorithm 3–4 for rounding and refinement by changing its constraint and initialization. And in the actual code, Algorithm 3–4 are combined as a single subroutine.

---

**Algorithm 2** Optimization procedure for the Locale algorithm

---

1: **procedure** LOCALEEMBEDDINGS(Graph $G$, Partition $P$)
2:     Initialize $V$ with $v_i = e(i)$, $i = 1, \ldots, n$.
3:     Initialize the ring queue $R$ with indices $i = 1, \ldots, n$.
4:     Let $z = \sum_{j=1}^{n} d_j v_j$.
5:     **while** not yet converged **do**
6:         $i = R.pop()$                     ▷ Pick an index from the ring queue
7:         $\nabla Q(v_i) = \sum_{j \in P(i)} a_{ij} v_j - \frac{d_i}{2m}(z - d_i v_i)$    ▷ Sums only $j$ in the same partition of $i$
8:         $g_i = \begin{cases} e(t) \text{ with the max } (\nabla Q(v_i))_t, & \text{if } \nabla Q(v_i) \leq 0, \\ \text{top}_k^+(\nabla Q(v_i)), & \text{otherwise.} \end{cases}$
9:         $v_i^{\text{old}} = v_i$,   $v_i = g_i / \|g_i\|$              ▷ Perform the closed-form update
10:        $z = z + d_i(v_i - v_i^{\text{old}})$                   ▷ Maintain the $z$
11:         Push all neighbors $j$ with nonzero $a_{ij}$ into the ring queue $R$ if it is not already inside.
12:     **end while**
13:     **return** the embedding $V$
14: **end procedure**

---

---

**Algorithm 3** Rounding procedure for the Locale algorithm

---

1: **procedure** LOCALEROUNDING(Graph $G$, Partition $P$, Embedding $E$)
2:     Initialize $V$ with input $E$.
3:     Run line 3–12 of Algorithm 2 with cardinality constraint $k = 1$.
4:     Let the index of the 1-sparse embedding above be the new partition $P'$.
5:     **return** $P'$
6: **end procedure**

---

---

**Algorithm 4** Refine and Aggregate procedure from the Leiden algorithm

---

1: **procedure** LEIDENREFINEAGGREGATE(Graph $G$, Partition $P$)
2:     Refine $P' \leftarrow$ *LocaleRounding*$(G, P)$ by restricting the local move within its partition.[5]
3:     Forms a hypergraph $G'$ by merging nodes inside the same partitions in $P'$ and simplify $P'$.
4:     done $\leftarrow |P|$ equals $|G'|$.
5:     **return** $G'$, $P'$, done
6: **end procedure**

## Footnotes

[4]Note that in Proposition 1, when $q \le 0$ and there are multiple maximum $q_t$, we further select the $t$ with the maximum $(v_i)_t$ in the previous iteration. This makes the limit to hold on the corner case $q = 0$.

[5]This is the refinement step implemented in the package `python-leiden`.