[Reviews · NeurIPS 2020]

Review 1

Summary and Contributions: The paper presents a new local-search algorithm for the problem of community detection in social networks under the modularity objective. The idea to relax the discrete step in the local-search process with a continuous-type step in a space represented by low-cardinality vectors. This gives a continuous solution which is then used as an initialization for a standard local-search phase. The proposed method is more efficient than a full-blown SDP relaxation, while improving empirically the performance of state-of-the-art local-search methods.

Strengths: 1. The problem of community detection in social networks (or other graphs) is extensively studied and relevant to the NeurIPS community. 2. The proposed method, including the local-search step and the rounding step, is quite interesting and, to my knowledge, original. 3. The empirical evaluation shows that the method improves significantly the state-of-the-art local-search algorithms.

Weaknesses: 1. The main weakness is that the proposed method is only a heuristic and there is no theoretical analysis to support the good empirical results.

Correctness: To the best of my understanding the claims and the method are correct. The empirical methodology is sound and presents a convincing case.

Clarity: The paper is relatively well-written.

Relation to Prior Work: To my knowledge the related work is discussed adequately, and the contributions are clearly identified.

Reproducibility: Yes

Additional Feedback: Post rebuttal: I thank the reviewers for their response.


Review 2

Summary and Contributions: The paper considers the problem of partitioning the nodes of a graph in k sets to maximize the number of edges that are between the vertices of the same set. The author call it the modularity maximization problem. Similar problems such as correlation clustering etc have been studied in the literature. The paper gives a higher dimensional version of the local search algorithm and shows, empirically, the bound it constructs are close to bounds from SDP. Moreover, the solution returned is able to escape local minima that are usually found in local search procedures that have been studied for the problem. The authors also show connections to low rank SDPs etc.

Strengths: The authors give a new (non-convex) relaxation for the problem and show that it can be solved (to a local maxima) faster than solving the SDP but, empirically, gives similar results. Moreover, these solutions can be used to obtain feasible solutions of the problem. The main novelty is the new non-convex relaxation and its empirical usefulness.

Weaknesses: There is no theoretical justification for the approach. While there is a new formulation that is considered in the paper, the algorithms is still too close to local search and the only gain from the approach is the initialization of the local search. A fair comparison could be to give many random initializations for the local search approach as can be afforded in the same time as the approach in this paper. For time limits, it compares itself to SDP solvers which is, of course, an easy target to beat.

Correctness: Yes.

Clarity: The writing of the paper is not clear and some of the claims and algorithms are hard to decipher. 1. The notation card(v) is weird. Maybe use |.|_0 (the zero norm) or |support(v)|. 2. The paper uses algorithms from previous works but never defines them. Since they are simple local search methods, it might be better to precisely define them.

Relation to Prior Work: Yes

Reproducibility: Yes

Additional Feedback: Thanks for the feedback. I updated the score based on the comments received.


Review 3

Summary and Contributions: Modularity is a measure used to quantify the quality of the partition of a graph (i.e., its node set) in k communities. It is a popular measure in practice despite its known shortcomings, including the resolution limits discussed by Fortunato and Barthelemy "Resolution limit in community detection." (PNAS). This paper's key contribution is an efficient solver for an SDP relaxation of the modularity objective. It consists of two steps, obtaining a "k-sparse" assignment matrix, that then is rounded to a community assignment. The proposed algorithm is projected gradient descent. The key results are stated as proposition 1 and theorem 1, on page 4, and their proofs have been provided in the appendix. I have been able to go through the proofs, and they are correct to the best of my knowledge. Proposition 1 that focuses on the subproblem for a specific node can be seen as a greedy soft community assignment. Figure 1 nicely illustrates in a toy example the benefits derived from their approach. From an experimental perspective, the author(s) has/have implemented their method, and tested it on some real-world datasets. The main baselines are the Louvain and Leiden methods. The latter improves Louvain algorithm with respect to ensuring communities are well connected. [Update: I thank the authors for their feedback.]

Strengths: - The reviewer believes that the main strength of the proposed method is that it improves in a non-trivial, yet highly practical way a family of heuristics that are used by many practitioners. - The paper is grounded in solid theory - The low cardinality solver comes with solid theoretical guarantees, at least on its convergence.

Weaknesses: - The reviewer believes that the main weakness of the paper is the lack of breadth of the contribution. Yes, I agree that modularity is popular as a measure, but at the end of the day the goal is to detect real-world communities better. The author(s) has/have narrowed down their contribution explicitly to improving Louvain and Leiden methods. It would have been more interesting to see precision/recall on communities with ground truth data (e.g., from SNAP), and use state-of-the-art approaches such as motif-aware/higher-order community detection. - Concrete scalability analysis is lacking. - While there is some novelty in the paper, it is limited from a technical perspective.

Correctness: Yes.

Clarity: Yes, it was a pleasure to read.

Relation to Prior Work: Yes.

Reproducibility: Yes

Additional Feedback: - Does the solver work well on overlapping community detection? Using the overlapping stochastic model, the authors could try to see how well they can recover the overlap.


Review 4

Summary and Contributions: The paper gives an improved algorithm for community detection using modularity maximization. It introduces a new technique based on semidefinite programming, that can be used in combination with current greedy methods. It is evaluated on five networks, two of which have over a million nodes. The algorithm runs well, and improves on the final modularity score

Strengths: The technique using SDP is quite nice and natural. The paper is technically pretty strong. Modularity is a very commonly used method for clustering, so this method would be quite useful

Weaknesses: The presentation is confusing at times, and could be improved. There are no worst case guarantees on how well the method performs. Some relevant work is not discussed

Correctness: Appears correct

Clarity: Could be improved

Relation to Prior Work: There is more work on modularity approximation using SDPs, e.g., [Dasgupta et al, JCSS 2012], which gives an approximation algorithm in some settings. That should be discussed. Also, there has been a lot of work on the limitations of modularity, e.g., [Good et al., 2010] that maximizing modularity is sometimes not the best in many networks. It would be good to mention this work, and place the paper within that context

Reproducibility: Yes

Additional Feedback: page 4: Q(v_i) is not defined, and not completely clear from the context, since Q(V) involves summation over terms i,j, which are in V. While its definition can be figured out by looking at the proof in the Appendix, it should be defined here The statement of Proposition 1 is pretty complicated. Some intuitive explanation would be helpful The discussion about k in section 3 is confusing. How does the time depend on k? Why is small k important? For modularity clustering, there is no a prior bound on the number of clusters, which is often considered as a good feature of the approach Caption of Fig 1: "indice"-->"index". The notion of index and value is not very clear from the discussion so far, and should be clarified, using the graph in the figure The subroutines LocaleEmbeddings, LocaleRounding and LeidenRefineAggregate used in Algorithm 1 are defined informally, and in a somewhat confusing manner in the previous subsections. It would be useful to have succinct pseudo-code for them as well Fig 2: does optimal value mean the maximum modularity score? The absolute difference in objective value is a bit hard to interpret. Might be useful to consider the ratio It is nice that the proposed algorithm gives better solutions with higher modularity score. But it would be useful to see whether the resulting clusters are better in some qualitative sense -------------------------------- I have read the author feedback, and feel they have addressed some of the comments

[Author Response · NeurIPS 2020]

We thank the reviewers for their thoughtful feedback, as well as the pointers to related work. We will address a few of the major comments below, and will incorporate these points and discussions into the revised paper.

First and foremost, we want to clarify some misunderstandings in Reviewer 2's comments on the paper, specifically the statements that "the algorithms is still too close to local search and the only gain from the approach is the initialization of the local search", and that "a fair comparison could be to give many random initializations for the local search approach as can be afforded in the same time as the approach in this paper."

We want to emphasize that this comparison is precisely what we did in the evaluation (we realize this can be stated more clearly and we will absolutely edit the paper to emphasize this point). Specifically, for the results in Table 2, and as stated in lines 263–268 of the text, the results we report are for Louvain/Leiden with 10 random restarts (the best way to improve their performance given the additional time), whereas the Locale entries use only a single initialization. Thus, the time taken for Louvain/Leiden in that comparison is *more* than the time using the Locale initialization, and Locale still performs better. While it is true that we also compare to semidefinite solvers, the above comparison is the practical point of emphasis when it comes to the speed of the method. We agree that the "10 iterations" (which counts the number of computing embeddings, rounding, and refining) vs. "10 trials" (which counts the number of random restarts) is a bit confusing upon re-reading, and we will absolutely clarify this in the resubmission, plus explicitly show time/performance curves for the different methods. We hope that this addresses Reviewer 2's main concern with the method, ensuring that we are indeed conducting an apples to apples comparison, where the Locale approach brings substantial benefits.

We next want to address several other comments made by the reviewers.

**R2 and R4** *(on including pseudo-code of the mentioned algorithm)*: Thanks for the suggestion. We agree that this will help to make the paper more self-contained and can certainly add this.

**R3** *(on comparisons to other community detection approaches)*: You are absolutely correct that the method and analysis here definitely do focus on improving the modularity optimization of Louvain/Leiden. While we completely agree that the real goal is the general community detection, and modularity is a single approach for this task, the methods are popular and ubiquitous enough that we believe a substantial improvement on them to be an important contribution to the literature (and so we focus largely on this improvement). However, we're definitely happy to provide additional experiments on communities with ground truth data from SNAP with a brief comparison to higher-order methods.

**R4** *(on the discussion about $k$)*: Sorry for the confusion on this point. We do not assume that we know the number of clusters overall (denoted by symbol $r$). Instead, the symbol $k$ in section 3 denotes the cardinality of candidate memberships for *each vertex* (i.e., this would be $k = 1$ for the greedy local move of Louvain/Leiden, even though they can of course discover many clusters). We observe that a small $k$ is sufficient to reach the global optimum of the SDP, but larger $k$ is also fine. However, since the Locale algorithm has a time complexity of $O(\mathrm{card}(A)klogk)$, we prefer a smaller $k$ as long as it is sufficient to satisfactorily optimize the SDP. We will clarify this in the paper.

**R4** *(on Fig 2)*: Yes, the optimal value in Figure 2 means the modularity score. We appreciate the suggestion and will change it to the relative ratio.

**R4** *(on additional references mentioned in the review)*: Thanks for the references! We will absolutely discuss them in the paper.

**R2, R3, and R4**: We appreciate the feedback on writing and presentation and will incorporate these suggestions into the paper.

[Meta-Review · NeurIPS 2020]

The authors introduce a new local search algorithm for community detection in social networks. The main strength of the paper is the strong empirical performances of the introduced algorithm. Its main limitation is the lack of theoretical guarantees for the new model.